# Sparse Winning Tickets are Data-Efficient Image Recognizers

**Mukund Varma T**[1], **Xuxi Chen**[2], **Zhenyu Zhang**[2], **Tianlong Chen**[2],
**Subhashini Venugopalan**[3], **Zhangyang Wang**[2]
[1]Indian Institute of Technology Madras, [2]University of Texas at Austin, [3]Google
mukundvarmat@gmail.com, vsubhashini@google.com
{xxchen,zhenyu.zhang,tianlong.chen,atlaswang}@utexas.edu

## Abstract

Improving the performance of deep networks in data-limited regimes has warranted much attention. In this work, we empirically show that "winning tickets" (small subnetworks) obtained via magnitude pruning based on the lottery ticket hypothesis [1], apart from being sparse are also effective recognizers in data-limited regimes. Based on extensive experiments, we find that in low data regimes (datasets of 50-100 examples per class), sparse winning tickets substantially outperform the original dense networks. This approach, when combined with augmentations or fine-tuning from a self-supervised backbone network, shows further improvements in performance by as much as 16% (absolute) on low sample datasets and long-tailed classification. Further, sparse winning tickets are more robust to synthetic noise and distribution shifts compared to their dense counterparts. Our analysis of winning tickets on small datasets indicates that, though sparse, the networks retain density in the initial layers and their representations are more generalizable. Code is available at `https://github.com/VITA-Group/DataEfficientLTH`.

## 1 Introduction

Deep convolutional networks [2] have achieved wide success on a variety of tasks, but they do demand large amounts of data. However, there exist various domains and tasks where training data - labeled or unlabeled - are limited. Often, in such data-limited regimes, transfer learning has emerged as a dominant approach [3] as it achieves superior performance compared to training from scratch with random initialization [4]. In some domains, the highly specialized training performant models can still be a challenge, *e.g.* in scientific or medical images [5], or just images from a different distribution - differing vastly in size, color, or channels - than seen in typical image datasets. In such cases, training large-capacity data-hungry networks might be less performant than training models of similar architecture and lower capacity [6]. While much of the literature on sparser and lower capacity networks has focused more on efficient models and mobile deployment [7], it is also known [8] that sparsity as a regularization could reduce overfitting. Thus it is interesting to study sparse networks in the context of low data or data efficient regimes (Fig. 1).

One of the popular ways of obtaining a sparse network with similar architecture but lower capacity is pruning [9]. Frankle et. al. [1] formulated the lottery ticket hypothesis (LTH) and proposed to use iterative magnitude-based pruning (IMP). They demonstrated that dense networks can be magnitude pruned to obtain highly sparse sub-networks ("winning tickets") which can match or potentially outperform the original dense network. The caveat is that the winning ticket needs to use the same random initialization as the original dense network. This approach also induces an inductive bias specific to the task to be learned, which leads to a better network when compared to a dense network of similar size trained from scratch [10]. Further, such pruning was recently found to reduce sample complexity in theory and practice [11]. It has also been suggested that the number of samples required to achieve zero generalization error is proportional to the number of the non-pruned weights in the

hidden layer [11]. However, despite implications of sparse networks being suitable for low data regimes, only limited recent works [12, 13, 14] have validated them practically, and no prior work has evaluated their effectiveness for image recognition tasks in the low data setting.

In this work, we provide empirical evidence to endorse the suitability of sparse networks, identified by IMP, for data-limited image recognition. We conduct extensive experiments to compare the relationship between network capacity, training data size, and image classification accuracy. We first show that sparse winning tickets have substantially better performance on low (50-100 samples per class) data regimes compared to their original dense counterparts. Next, we show that sparse winning tickets for data-efficient learning compliments and can be combined with existing methods such as augmentations or fine-tuning (from a self-supervised backbone), to further boost their attainable performance by as much as 16%. We further assess the generalization capacity of the networks through detailed robustness tests on synthetic and domain-shifted datasets. We also evaluate special data-deficient cases like long-tailed classification to compare performance when some classes have far few samples than others. Finally, we try to understand the properties of sparse networks that lead to their superior performance. We find that both capacity and connectivity together are important for generalization. Our key contributions can be summarized as:

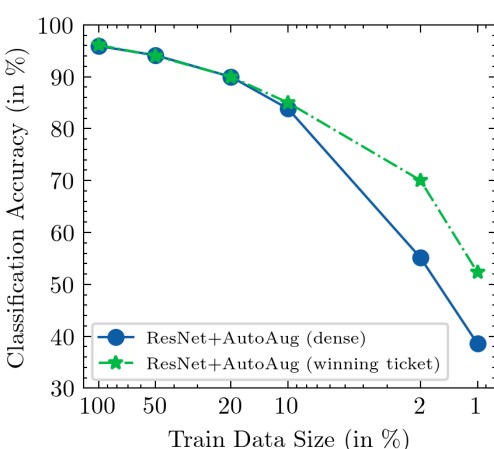

Figure 1: Sparse winning tickets are data efficient recognizers. Winning tickets (green) obtained by iterative magnitude pruning are not only sparser but also show considerably improved performance, by as much as 16% (absolute), compared to the original network (blue) when trained on smaller subsets (as little as 1% size) of CIFAR10 dataset.

1. We find that the sparsity of the winning ticket increases with a reduction in training data.
2. We demonstrate that winning tickets can easily combine with existing data-efficient training strategies such as augmentations to outperform similarly trained dense networks on low data regimes by as much as 16%.
3. We further show that winning tickets, on low data regimes, are robust to synthetic noise and domain shifts, and show improved performance on long-tailed classification.
4. Our analysis indicates that on data-limited regimes, sparse winning tickets continue to preserve density in initial layers, and both capacity and connectivity are important for generalization.

## 2 Related Works

**Sample-Efficient Image Recognition.** Learning from few samples and less data has been a long-standing topic of interest in machine learning. While transfer learning, particularly from ImageNet pretrained models [3] is the most popular approach, semi-supervised techniques [15] and self-supervision [16] have been successful when there is an abundance of unlabeled data in the same domain. All of these typically involve pretraining the model on data from a similar domain, or the same domain, often with a variety of augmentations and then tuning it on a few labeled examples. These are complimentary to our approach, and we explore combining augmentations and transferring in our experiments. Few-shot methods are also related and they usually look at tuning just the final classification layer [5, 17]. A different approach is meta-learning [18], which aims to identify a network from the support classes, seen during pre-training, and which can easily adapt to the target set of classes in few steps. Random pruning was mentioned for few-shot learning [19] , but much was left for future investigation. Our work is different from typical few-shot methods in two ways: ① We do not limit ourselves to transfer learning and also focus on training from scratch directly from the small dataset; ② We cover data sets with approximately 50-100 samples per class which lies between most few shot settings and typical deep learning settings with more data.

There are also works that directly use small datasets without external dependence on pre-training, often relying on training strategies to prevent overfitting. Some researchers propose to use the cosine loss [20] while some use the t-Distributed von Mises-Fisher (t-vMF) loss [21] along with the standard cross entropy for training. These regularizing loss functions reduce intra-class variance. Some other methods rely on extracting useful geometric priors to help avoid over-fitting, such as preset cosine filters [22] and full convolutions [23] which exploits the absolute spatial location of objects in the image and improves translation invariance to strengthen the visual inductive prior. In this work, we cover datasets containing roughly 50-100 samples per class and the above methods have shown strong performance in this regime [24] and offer easy implementation. Therefore, in this work we compare against these methods and study if they supplement pruning techniques.

**Pruning.** Among pruning based methods, the iterative magnitude pruning (IMP) (described in Sec. 3.1) [25] is most relevant to our work. IMP is a process that helps identify sparse sub-networks ("winning tickets") within a larger dense network that is capable of matching or potentially out-performing the dense network. This has been successful on a variety of tasks including image classification [26], natural language processing [27, 28], and generative modelling [29]. While IMP has been popular for obtaining sparse networks, we are not aware of works that explore its relationship to data efficiency. In this paper, we carry out detailed experiments to validate if sparsity, particularly "winning tickets" obtained via IMP, improves data-efficiency of deep neural networks.

## 3 Methodology and Experiments

Our approach primarily uses iterative magnitude pruning (IMP) and we study the performance of the "winning tickets" in low data regimes. In this section, we describe IMP, and provide an overview of our experimental studies including the basic experimental setup, augmentation strategies we compare against, and the datasets used for different studies.

### 3.1 Preliminary: Iterative Magnitude Pruning (IMP)

IMP [25, 1] is introduced as a way to identify sparse sub-networks within a dense network that match or exceed the performance of the full dense network. They view these sub-networks as having won the initialization "lottery" and refer to them as "winning tickets". We use the IMP procedure with rewinding [30, 31] and describe it briefly below.

Let $f(\mathbf{x}, \boldsymbol{\theta})$ denote the neural network, where $\boldsymbol{\theta}$ denotes its parameters and $\mathbf{x}$ denotes its input. Then the sub-network can be characterized by a binary mask $\mathbf{m}$ and can be applied to the original network via a Hadamard product ($\odot$) to obtain the sub-network $f(\mathbf{x}, \boldsymbol{\theta} \odot \mathbf{m})$. For a network initialized with $\boldsymbol{\theta_0}$, the IMP process aims to find a final mask $\mathbf{m}$ of a specified sparsity ratio ($s$) as follows:

1. initialize $\mathbf{m}$ to be all ones mask.
2. train $f(\mathbf{x}, \boldsymbol{\theta_0} \odot \mathbf{m})$ for $r$ steps to get $\boldsymbol{\theta_r}$. This will be used as the initialization for the later pruning iterations.
3. continue to train $f(\mathbf{x}, \boldsymbol{\theta_r} \odot \mathbf{m})$ to convergence.
4. remove a portion of the weights (determined by the pruning rate $p$) with the smallest magnitudes from $\boldsymbol{\theta} \odot \mathbf{m}$. Update $\mathbf{m}$ by setting the removed weights to 0.
5. repeat steps 3, 4 until the specified sparsity ratio ($s$) of pruned vs. original weights is achieved.

Note that when $r = 0$, the above algorithm reduces to IMP without rewinding. Rewinding is found to be essential for the successful and stable identification of winning tickets in large networks [30, 31]. Since this is an iterative process, we obtain sub-networks of different sparsity at each stage. In all our experiments, the sub-network that has the best performance, across all iterations, on the validation set is termed as the "sparse winning ticket".

### 3.2 Overview of Experimental Studies

We provide a broad overview of our experiments before presenting the details and datasets used.

(1) We first study how the network capacity of models pruned via IMP affects classification performance when training on datasets of different sizes. Since augmentations are often employed during the training of models in low-data regimes, we also perform this study with models trained

using different augmentation strategies. We compare the performance of dense models with just augmentations against those with both augmentation and pruning.

(2) Next, to study if sparsity prevents over-fitting by avoiding memorization, we evaluate the performance of the pruned models on a synthetically transformed dataset and a domain-shifted dataset.

(3) Aside from augmentations, we also compare IMP against other training techniques employed in low-data regimes as discussed in Sec. 2, as well as fine-tuning from ImageNet and SimCLR backbones. All our main experiments are on ResNet-18, and we also show these results hold on ResNet-50, non-residual networks like VGG, and parameter-efficient MobileNet-v2.

(4) To verify the generalizability of the approach and measure performance of sparse winning tickets in data-limited regimes, we perform studies on (a) 3 real datasets with few (50-80) images per class, and also (b) simulated datasets with imbalanced (long-tail of) classes. We also present results on more complex data subsets of ImageNet and CIFAR100 with only 5-50 images per class.

(5) Finally, we try to understand why sparse winning tickets are able to perform well in data-limited regimes. We do this by analyzing network capacity and connectivity, and also how pruning affects the layers - which layers are getting pruned, the layer-wise representation similarity, and the generalizability of the representations.

### 3.3 Experimental Setup

We use the ResNet model (specifically ResNet-18 unless otherwise specified) in all our experiments. Since we are working with smaller-sized images (often 32x32 or 64x64), we modify the initial convolution layer to a 3x3 kernel, with padding 1 with no max-pooling. For images of size 224 we do not make any modifications. We follow the IMP lottery ticket-finding procedure with 16 pruning iterations where $20\%$ of the weights are pruned after each iteration. In each pruning iteration, the model is trained for 200 epochs to minimize the cross entropy loss using the SGD optimizer with weight decay 0.0005. The initial learning rate is set to 0.1 and then cosine decayed for 200 epochs. For fine-tuning experiments, we use a lower learning rate of 0.001. During the first iteration, we use the rewinding technique and set $r = 2$ epochs.

### 3.4 Augmentation Strategies

Augmentations are one of the default strategies applied when working with limited data. In this work, we look at augmentations in two scenarios: (1) as a comparison to the performance of the winning ticket, *i.e.*, comparing the dense model with augmentations to that of a pruned model without augmentations, and (2) we combine augmentations with IMP to see the benefit of augmentation in addition to pruning. We carry out experiments with four different augmentation strategies each of different strengths. The strategy Basic consists of just RandomCrop and RandomFlip. All the other strategies are composed on top of the Basic augmentation. Contrast strategy additionally consists of ColorJitter and RandomGray [16]. Auto uses Basic augmentations and also the best policies identified for each dataset by AutoAugment [32]. Rand is based on RandAugment [33], and composes four randomly selected augmentations in addition to the Basic augmentations. We can potentially order these 4 augmentation strategies by increasing strength as: Basic < Contrast ≈ Auto < Rand.

### 3.5 Datasets

**CIFAR10** [34] is our primary dataset for all analysis experiments. It consists of 60,000 colour images (train: 50,000, val: 10,000) of size 32x32 split into 10 classes. For experimental comparisons to the data-limited regime, we sub-sample it at various sizes while maintaining the original class-size ratios. E.g., at $1\%$ data size, the training set contains 50 images per class. We aways evaluate on the full validation set in all our experiments. For the robustness and extended experiments, we use additional datasets described in the corresponding sections.

**CIFAR10-C** [35]. We use the CIFAR10-C (C for corrupted) dataset to evaluate the robustness of the models to (unseen) **synthetic transforms**. It contains 19 different transforms each of 5 different strength levels. These transforms can be broadly classified into - Noise (e.g. impulse noise), Blur (e.g. motion blur), Weather (e.g. snow), Digital (e.g. JPEG), and Extra (e.g. brightness).

**CIFAR10.2** [36]. To evaluate the network's performance on **domain shifts**, we use the CIFAR10.2 dataset derived from TinyImages [37]. It contains the exact same classes as the original CIFAR10 dataset but has a different and harder collection of images ideal for evaluating distributional shifts.

**CLaMM** [38], **ISIC** [39] and **EuroSAT** [40]. Aside from experiments on subsampled versions of CIFAR, we use three diverse low-data regime datasets covering different domains with varying numbers of classes and images per class. The EuroSAT dataset [40] contains $64 \times 64$ sized satellite images from 10 classes with 50 images per class captured from Sentinel-2 satellite. It consists of multiple channels. We follow one standard procedure [40] to use the RGB channels only. ISIC 2018 dataset [39] contains $224 \times 224$ sized dermoscopic skin lesion images from 7 classes with 80 samples per class. The CLaMM dataset [38] contains $224 \times 224$ sized gray-scale latin scripts from hand-written books from 12 classes with 50 images per class.

**CIFAR100** [34] For studies on the performance of the sparse networks on **imabalanced and long-tailed classification** tasks we derive a dataset based on CIFAR100. We selectively sample the CIFAR100 dataset such that the class-wise image count follows a long-tailed distribution (defined by an exponential curve given by: $\frac{N}{n} \times \lambda^{j/(n-1)}$ where $N$ is total number of samples in the dataset, $j$ is the class index and $n(=100)$ is the number of classes in the dataset. The degree of "long-tailedness" is defined by the imbalance factor ($\lambda$), i.e a lower value indicates more classes have a lower sample count. The derived long-tailed class-wise distribution of CIFAR100 is visualized in the Appendix C.

## 4 Results

### 4.1 Sparse winning tickets show superior performance in low-data regimes.

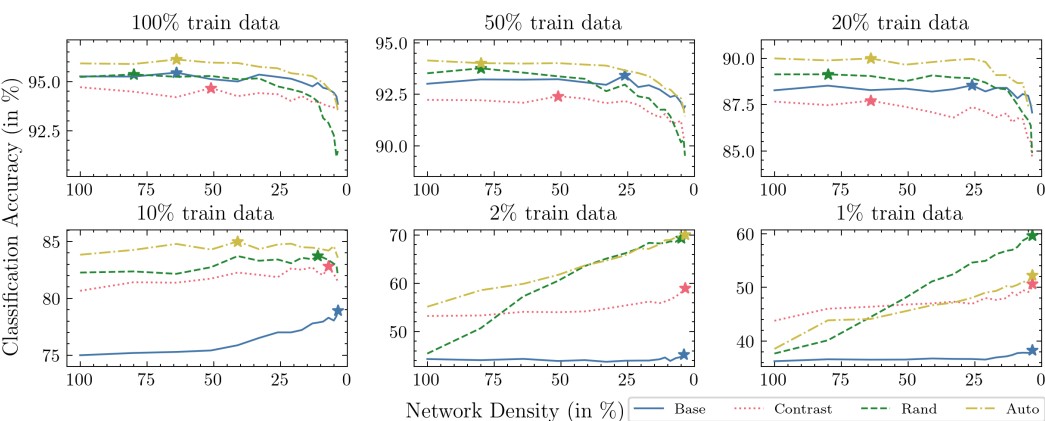

Figure 2: Combining pruning with augmentation strategies on CIFAR10. Performance on data subsets of different sizes (subplots), at different sparsity levels (x-axis) for 4 augmentation strategies (lines). $\star$ indicates winning tickets. Winning tickets are sparser and substantially better performing at low-data regimes (row 2). Augmentations further improve the performance of the winning tickets.

We first verify our main claim and show the suitability of sparse winning tickets for data-limited regimes. Fig. 3a (and Fig. 1) presents the results of training a ResNet-18 network on varying CIFAR10 subsets of sizes $100\%$, $50\%$, $20\%$, $10\%$, $2\%$, and $1\%$. At the lower data sizes, the sparse winning ticket (denoted as **WT**) shows considerably improved performance compared to the dense model.

We also investigate how pruning compares with augmentation strategies and if both of these can be combined. Fig. 3a shows the results of the best-performing dense and sparse networks with augmentation. The best is chosen across all augmentations for each data size and are indicated by **Dense+Aug**$^*$ (or **D+Aug**$^*$) for the dense model and **WT+Aug**$^*$ for the winning ticket. We can see that across all data sizes, models pruned with augmentation outperform unpruned ones and as the data size decreases the gap between the two widens indicating that sparse networks are suitable for data-limited regimes.

Fig. 2 presents detailed comparisons of the performance of the model on the different data subsets (each subplot) at different sparsity levels (subplot x-axis) for the different augmentation strategies (lines of different styles). In almost all cases, the best performing model is sparse i.e density $< 100\%$. While in the higher data regimes (top row), there is a decrease in performance with increased sparsity, the trend is reversed in the low data regimes (bottom row). More importantly, at the lower data sizes i.e $2\%$, $1\%$, it's clear that the sparse winning tickets (indicated by stars in Fig. 2) significantly outperform the dense ($100\%$ density) models with augmentations. Further, among the augmentation strategies, Auto-augment seems to consistently yield the best performance, except in the $1\%$ data size where Rand-augment does much better. We further discuss the relevance of applying augmentations during the pruning stage in Appendix. A.

## 4.2 Winning tickets are robust to distributional shifts.

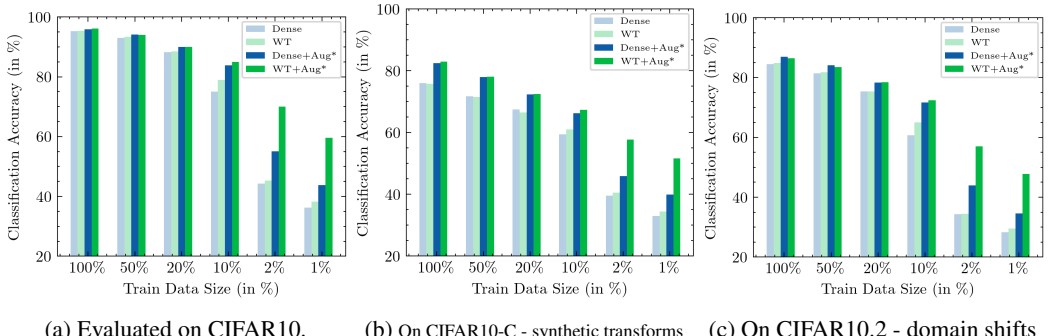

(a) Evaluated on CIFAR10.  (b) On CIFAR10-C - synthetic transforms  (c) On CIFAR10.2 - domain shifts

Figure 3: Comparison of the dense model, and sparse winning ticket (WT), and the best performing dense and sparse models with augmentation (+Aug*) trained on varying CIFAR10 subsets and evaluated on CIFAR10, and CIFAR10-C and CIFAR10.2 for robustness.

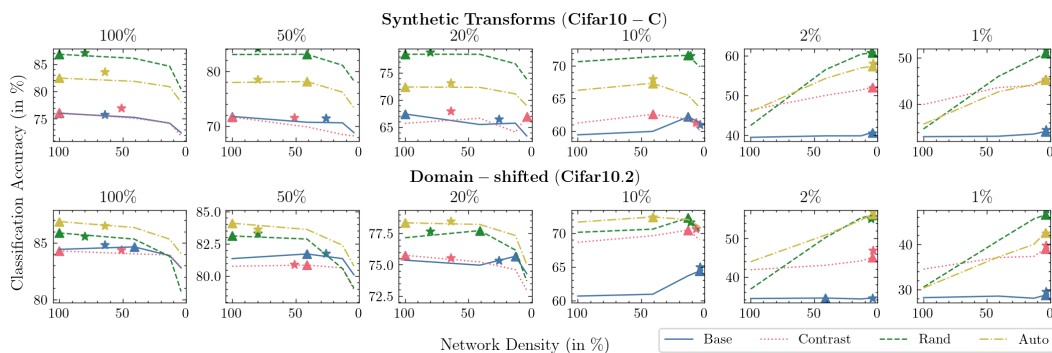

Figure 4: Robustness of models trained on CIFAR10 subsets on synthetic transforms (row-1) and domain-shifted (row-2) datasets. Shows performance of models trained on CIFAR10 subsets of different sizes (columns) at different sparsity levels (x-axis) for 4 augmentation strategies (lines). Sparse winning tickets identified on the CIFAR10 val. set are indicated by ⋆ and △ indicates the most robust models.

As the data size gets smaller, the chance of overfitting and memorization of the training samples increases. Thus, to study robustness, we evaluate the performance of the identified sparse winning tickets on two types of **distributional shifts**: (a) synthetic transforms via the CIFAR10-C dataset [35] and (b) domain-shifts via the CIFAR10.2 dataset [36]. Note that we only perform evaluation (i.e. inference) on these datasets to study the robustness of the winning tickets to these transformations. We evaluate the network's performance on:

**(a) synthetic transforms** on the CIFAR10-C dataset. Row-1 in Fig. 4 and Fig. 3b visualizes the average accuracy of the models across all 19 unseen transforms.

| METHOD | CIFAR10 (2%) | | CIFAR10 (1%) | | CLaMM | | ISIC | | EuroSAT | |
|---|---|---|---|---|---|---|---|---|---|---|
| | D+Aug* | WT+Aug* | D+Aug* | WT+Aug* | D+Aug* | WT+Aug* | D+Aug* | WT+Aug* | D+Aug* | WT+Aug* |
| RANDOM INIT. (R18) | 55.14% | 70.05% | 43.8% | 59.66% | 50.29% | 55.76% | 57.34% | 59.73% | 83.44% | 87.85% |
| IMAGENET INIT. (R18) | 75.50% | 77.92% | 66.00% | 69.46% | 47.46% | 55.86% | 59.72% | 62.80% | 90.75% | 91.32% |
| IMAGENET INIT. (R50) | - | - | - | - | 51.66% | 57.03% | 61.73% | 64.88% | 92.89% | 93.39% |
| SIMCLR INIT. (R18) | 52.58% | 64.09% | 37.56% | 44.39% | - | - | - | - | - | - |
| COSINE LOSS | 64.82% | 72.63% | 45.87% | 64.67% | 49.60% | 60.15% | 59.03% | 61.00% | 82.28% | 88.68% |
| T-VMF LOSS | 62.23% | 72.81% | 41.70% | 64.54% | 24.50% | 59.67% | 56.22% | 59.30% | 71.14% | 88.26% |
| FULL CONV. | 62.16% | 73.24% | 49.30% | 64.20% | 46.77% | 57.91% | 56.42% | 58.20% | 76.05% | 77.06% |
| HARMONIC NETS | 61.36% | 66.48% | 22.97% | 49.85% | 42.58% | 44.14% | 50.56% | 52.51% | 79.40% | 83.28% |
| VGG | 64.64% | 71.01% | 51.8% | 61.71% | - | - | - | - | - | - |
| MOBILENETV2 | 64.64% | 71.01% | 51.8% | 61.63% | - | - | - | - | - | - |

Table 1: Comparing the dense model (D+Aug*) and winning ticket (WT+Aug*) using IMP in combination with other data-efficient training techniques on (columns 2, 3) 2% and 1% data subsets of CIFAR10, (columns 4, 5, 6) diverse datasets with few examples. R18 and R50 denote ResNet-18 and ResNet-50. The sparse winning ticket substantially outperforms dense model with all approaches.

**(b) domain shifted** CIFAR10.2 dataset consisting of wider variety of harder images from the CIFAR10 classes. Row-2 in Fig. 4 and Fig. 3b presents the results.

In both cases, similar to performance on the CIFAR10 data (Fig. 2) on 10% and lower training data sizes, sparser winning tickets generalize better to these unseen transforms and distribution shifted data, much more so with augmentations. The difference in classification accuracy between the sparse winning ticket and dense model is also more clearly noted from Figures. 3b and 3c where we look at the just the best performing models (as determined on CIFAR10 val. set) - dense model and winning tickets with and without augmentations. These results clearly indicate that sparse winning tickets, when trained as typical with augmentations, are capable of avoiding memorization of training samples at the smaller data sizes, and much more so than the dense models. This reinforces its suitability for data-limited regimes.

### 4.3 IMP complements existing data-efficient training.

Table. 1 (columns CIFAR10 2%, 1%) summarizes results using IMP to supplement and complement existing data-efficient methods. In particular, we apply IMP in combination with methods that have been shown to work well on small datasets: fine-tuning from ImageNet and SimCLR [16] backbones, self-regularizing cosine loss [20], use of full convolutions [23] (Full Conv.), as well as 2 emergent methods, t-VMF loss [21] and harmonic nets [22]. Networks are trained using the best-identified augmentation strategy at each data size i.e Auto-augment at 2% and Rand-augment at 1% data sizes respectively. Despite the simplicity of our approach, the winning ticket identified from a randomly initialized network outperforms almost all specialized data-efficient methods. When combined with the above data-efficient techniques, IMP further improves performance on the CIFAR10 val. set on an average by 8% and 15% at 2% and 1% data sizes respectively. IMP shows more significant gains with non-fine-tuning methods, as (absolute) magnitude-based pruning strategy does not work well with pre-trained initializations [41], though we do see substantial improvements even on fine-tuning. As seen in Table 1, our results on ResNet-18 extend beyond residual networks to VGG, and MobileNet-v2 based architectures.

### 4.4 Generalization to low sample datasets.

To evaluate generalization to datasets other than subsets of CIFAR10, we experiment on three diverse datasets. The EuroSAT [40], ISIC [39], and CLaMM [38] datasets cover different domains - satellite imagery, medical images, and books respectively with 7-12 classes but few images (50-80) per class. Table. 1 (columns CLaMM, ISIC, EuroSAT) compares the performance of the dense model and IMP, using the Basic augmentation strategy and in combination with existing data efficient techniques for low sample data. The winning tickets outperform the dense model on an average by 12%, 2.3%, and 5.5%

| DATASET | D+AUG | WT+AUG |
|---|---|---|
| IMAGENET (5%) | 28.82% | 31.04% |
| CIFAR100 (2%) | 17.21% | 25.06% |
| CIFAR100 (1%) | 11.21% | 16.44% |

Table 2: Comparison of the dense (D+Aug), and sparse winning ticket (WT+Aug) trained on complex data subsets with many classes.

on the CLaMM, ISIC and EuroSAT datasets respectively. Table. 2 discusses results on 2% (or 10 samples/class) and 1% (or 5 samples/class) data subsets of CIFAR100 and ImageNet 5% [42] (or 50 samples/class), and our results indicate the effectiveness of winning tickets to handle much larger number of unique classes even in the absence of sufficient data.

## 4.5 Long-tailed classification.

Long-tailed classification is a specific data-limited setting where some classes of the dataset contain very limited samples compared to the rest. The presence of the dominant class can lead to over-fitting. Therefore, we evaluate the effectiveness of sparse networks in such a dataset derived from CIFAR100 (see. Sec. 3.5). Table. 3 summarizes these results and we can see that the winning ticket has slightly better performance than the original dense network. An important note is that the total dataset size after imbalance is between $40\%$ and $20\%$ of the actual size (100-200 examples per class) therefore, the performance

| Imbalance Factor | D+Aug. | WT+Aug. |
|---|---|---|
| 0.10 | 64.77 | 64.92 |
| 0.05 | 58.43 | 59.43 |
| 0.02 | 50.91 | 51.72 |
| 0.01 | 45.13 | 46.73 |

Table 3: Comparison of the dense and sparse winning ticket with basic augmentations for long-tailed classification on data that's imbalanced to different extents.

improvements are smaller. Nevertheless, the gains indicate that sparse networks are able to avoid over-fitting due to class imbalance and not just uniformly reduced data size.

## 5 Discussion

Here we empirically investigate the factors that contribute to the winning ticket network's performance gains in data-limited regimes.

### 5.1 Network capacity and connectivity are important.

We first investigate if it is the winning network's capacity or the connectivity between neurons (that are retained after pruning) that contribute to the performance. During pruning even as the network capacity is reduced, the connectivity between layers is also changed. We study the original dense network, and the sparse winning ticket with 3 additional networks: (1) A small dense network which is a dense network but has overall parameters reduced to match the winning ticket. (2) A randomly pruned network where we take the original dense model, but randomly set some of the weights to zero to match the sparsity of the winning ticket. (3) A same-layer-sparse network that follows the exact same layer-wise densities as the winning ticket but the connectivity between the layers is random i.e. the binary mask at each layer is randomly initialized but must match the total number of active parameters in that specific layer as the winning ticket. We train these networks on the different CIFAR10 subset sizes using the identified best augmentation strategy and compare their performance in Fig. 5. The performance of the networks with lower

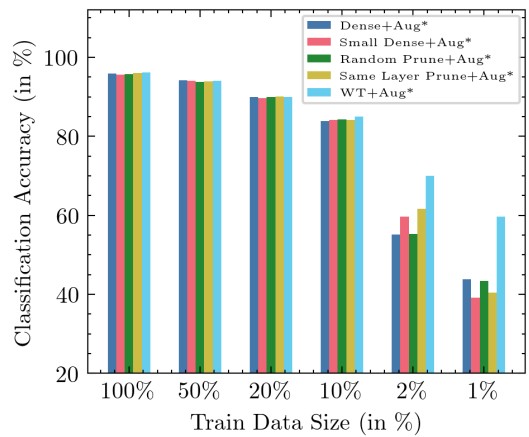

Figure 5: Comparison of the base dense, small dense, randomly pruned, same-layer-sparse, and winning ticket models trained on varying CIFAR10 data sizes.

capacities (small dense, random pruned, same-layer-sparse and winning ticket) are all improved compared to the base dense network particularly in the lowest data size setting, indicating that a smaller network size improves data-efficient performance. However, the winning tickets significantly outperform a network of similar capacity i.e. the "Small Dense", indicating that beyond capacity perhaps the network connections also play an important role. To probe the importance of connectivity, we compare the winning ticket (WT+Aug*) against randomly pruned (Random Prune + Aug*) and the same-layer-sparse network and observe that it outperforms both, emphasizing the importance of

connectivity. These lead us to conclude that both network capacity and connectivity play a vital role in improving the data efficiency of sparse networks.

## 5.2 Which layers are getting pruned?

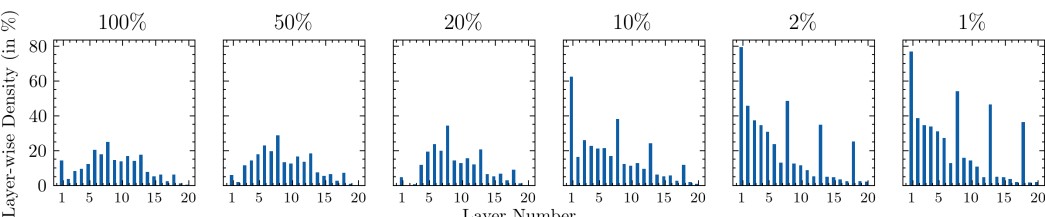

Figure 6: Layer-wise density of the sparsest winning tickets (i.e pruning iteration 16) for different training subsets. On smaller training data sizes, more weights from the initial layers are retained, and peaks appear at the residual connection layers of each block.

Fig. 6 presents the layer-wise density plots of the sparsest winning ticket at various training data sizes. Here density is defined as the ratio of the number of active parameters to the total number of parameters in that specific layer. We use the sparsest model identified (i.e $16^{th}$ pruning iteration) using the best augmentation strategy at that particular data size. We can see a striking difference between the sparse models identified at higher dataset sizes when compared to the lower ones. Particularly, the initial layers are denser as the training data size reduces. Retaining more weights in the initial layers appears to allow the sparse models to keep the filters for detecting primitive features such as edges, and corners perhaps helping it generalize better. Another interesting thing to note is that the identified sparse networks at lower data sizes have three peaks that correspond to the residual connection layers of each block. As discussed in Sec. 4.2, the winning tickets exhibit higher robustness to several corruptions, especially in the least data sizes (2%, 1%). We hypothesize that the retention of denser residual layers enables minimal change to the output even upon input change (or identity connections), hence enabling increased robustness [43].

## 5.3 Generalizability of the learned representations.

To empirically verify that the representations of the sparse winning ticket are more transferable we take 4 models: the dense network train on 100% of the data, along with the dense, dense+aug., and WT+aug. networks trained on 1% of CIFAR10. In Table. 4 we present the results of freezing the backbone and fine-tuning only the final linear classifier on the CIFAR100 dataset. Surprisingly the winning ticket not only outperforms the dense (1%), and dense+aug. network, but it also outperforms the dense model trained on 100% of CIFAR10. This indicates that the sparse sub-networks identified at the least data size learns generalizable representations of the input image.

| SUB-NETWORK | ACCURACY |
|---|---|
| REFERENCE (DENSE, 100%) | 20.79% |
| DENSE (1%) | 6.42% |
| DENSE+AUG. (1%) | 4.94% |
| WT+AUG. (1%) | 23.67% |

Table 4: Performance of linear classifiers on CIFAR100 for different sub-networks (backbones) trained on CIFAR10 data (% of data indicated in brackets) to evaluate generalizability of representations.

## 5.4 Layer-wise representation similarity

We use Centered Kernel Alignment (CKA) [44] to study the similarity of internal representation structure across different models in Fig. 7. CKA computes the normalized Hilbert-Schmidt independence criterion between two representations either within or across networks, averaged across samples within a mini-batch. It is better suited to measure similarity between high dimensional representations where the dimensions far exceed the number of data points. We visualize the intra-layer wise similarity across three models: dense network trained on 100% data, along with dense+Aug*, and WT+Aug* networks trained on 1% CIFAR10. We see that the sparse winning ticket exhibits a larger span of similar initial layers unlike the others which exhibit more intra-block similarity.

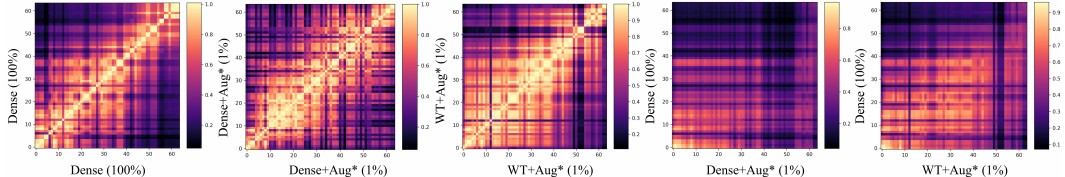

Figure 7: CKA Layer wise representation similarity across various networks. Comparing the similarities of the layers in the Dense, Dense+Aug*, and WT+Aug* with themselves, and then the Dense+Aug* and WT+Aug* with the Dense models.

It is possible that the increased layer-wise similarity is directly related to the denser residual connections observed in Sec. 5.2. To probe this, we plot the ratio of norms of the output from the residual branch and the main branch in Fig. 8. A higher value would indicate stronger propagation of information from the residual branch while a lower value would indicate the vice-versa. Surprisingly, these ratios are lower in the sparse network when compared to the dense network, which potentially suggests that the sparse network is in fact learning globally generalizable features. It is expected that a network trained on the complete 100% data is sufficiently more generalizable than one trained on fewer samples. Therefore, we compare the WT, Dense (+ Aug*) networks trained only on 1% data to a network trained on the complete 100% data, and we notice higher layer-wise similarity in the

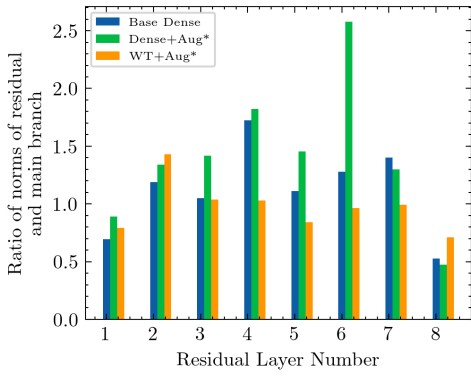

Figure 8: Ratio of norms of the output from the residual branch and the main branch for the network trained on 1% data subset.

case of the sparse network. These observations go in hand to further support our empirical analysis regarding the generalizability of the learned representations.

## 6 Conclusion

In this work, we hypothesized that winning tickets identified via magnitude pruning would be more effective in low-data regimes than their dense counterparts. Based on extensive experiments on sub-sampled versions of the CIFAR10 dataset, we find that with decreased training data, the winning ticket gets sparser and when combined with augmentations considerably outperforms the original dense network. We also verify that the sparsity of the winning tickets helps it avoid memorization and prevents over-fitting, by evaluating performance on distributionally shifted datasets. We show that IMP continues to show improvements in performance when combined with other data-efficient training strategies for low-data regimes. We further evaluate the generalizability of the approach on diverse datasets and simulated imbalanced datasets which have 50-100 examples per class. Finally, we analyze why sparse winning tickets show superior performance on low-data regimes. Our findings indicate that their performance can be attributed to combined properties of lower network capacity and connectivity, and help the network learn more generalizable representations.

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

## A    Augmentation is important to find better Winning Tickets

We can see from 3a that at 1% data size, the performance gap between WT+aug., Dense+aug. is significantly higher than their corresponding non-augmented versions. To verify the importance of augmentation during lottery ticket finding, we select the WT identified via Base-augment strategy, reinitialize the network to $\theta_r$ (refer Sec. 3.1) and train using the best augmentation strategy at each data size. From Table. 5, we can see that the winning ticket identified via Base-augment cannot match the performance of the WT+aug* even when trained later with the same best augmentation strategy (indicated by **WT(Base)+Aug***). Therefore, augmentation plays a significant role during ticket finding.

| DATA SIZE | WT | WT(BASE)+AUG* | WT+AUG* |
|---|---|---|---|
| 100% | 95.45% | 95.85% | 96.12% |
| 50% | 93.42% | 94.1% | 94.01% |
| 20% | 88.54% | 90.01% | 89.99% |
| 10% | 78.93% | 83.23% | 85.01% |
| 2% | 45.25% | 57.74% | 70.05% |
| 1% | 38.31% | 47.4% | 59.66% |

Table 5: Comparison of the winning tickets trained with Base-augment (WT), best augment (WT+aug*) and WT(Base) trained with best augment (WT(Base)+aug*) to evaluate the importance of augmentation during lottery ticket finding.

| DATA SIZE | RATIO |
|---|---|
| 100% | 1.07 |
| 50% | 1.09 |
| 20% | 1.22 |
| 10% | 1.33 |
| 2% | 1.39 |
| 1% | 1.23 |

Table 6: Average ratio of norms of the remaining weights from the winning ticket and corresponding dense network trained on CIFAR10 subsets.

## B    Adversarial Robustness of Winning Tickets

To evaluate the network's performance on adversarial attacks, we perform the FGSM [45] one-step attack with an epsilon value of $8/255$. Similar to Sec. 4.2, Fig. 9 presents adversarial robustness of models trained on CIFAR10 subsets across different augmentation strategies and sparsity levels. Unlike the other cases (i.e synthetic and domain shifts), the sparse networks perform worse than their corresponding dense counterparts across all data sizes. We hypothesize that since the IMP procedure only retains the maximum magnitude weights, they would directly affect the gradients, hence increasing the effects of the adversarial attack. To verify the same, we compute the average ratio of norms of the remaining weights between WT+aug. and Dense+aug across layers. Table. 6 clearly indicates that the winning tickets though sparse, have a higher magnitude of weights across all data sizes (ratio $> 1$). At the least data sizes (2%, 1%), the absolute value of the remaining weights is roughly 1.3 times its dense counterpart thereby increasing the magnitude of the adversarial attack for a given epsilon value.

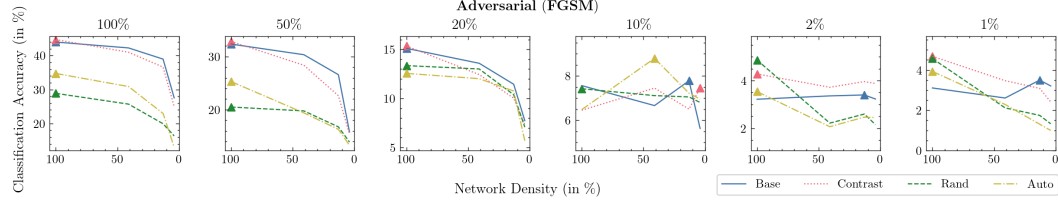

Figure 9: Robustness of models trained on CIFAR10 subsets on adversarial attacks. Shows performance of models trained on CIFAR10 subsets of different sizes (columns) at different sparsity levels (x-axis) for 4 augmentation strategies (lines). $\triangle$ indicates most robust models.

## C    Long-tailed CIFAR100 Data Distribution

We visualize the long-tailed data distribution of the sampled CIFAR100 dataset for varying $\lambda$ values in Fig. 10. The number of samples per class varies from 5-500 samples/class when $\lambda = 0.01$ up to 50-500 samples/class when $\lambda = 0.1$.

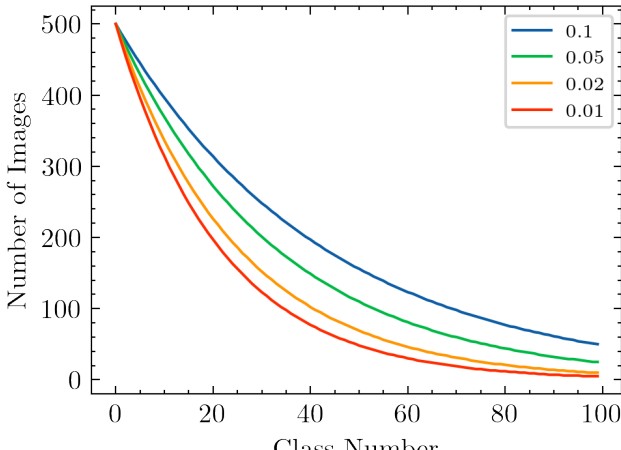

Figure 10: CIFAR100 long-tailed class-wise sample distribution across different imbalance factors $\lambda$.

