# OpenReview forum: "Sparse Winning Tickets are Data-Efficient Image Recognizers"
_NeurIPS.cc/2022/Conference — NeurIPS 2022 Accept_

### Official Review · Reviewer_Skos · 2022-07-06

**Rating:** 6
**Confidence:** 4
**Soundness:** 3 good
**Presentation:** 3 good
**Contribution:** 2 fair

**Summary:**

This paper shows that winning tickets can be successfully used in data-limited regimes. The authors conduct several experiments to empirically validate their assumptions. The authors first show that sparse winning tickets have superior performance in data-limited regimes, also compared to a dense network of similar size or a randomly pruned network. Then, they also show that they are robust to distributional shifts, such as domain shifts and synthetic transforms applied to the data. Most of the results are based on the CIFAR10 dataset, but they consider also other low sample datasets from various application fields, including cases with unbalanced classes.

**Questions:**

- Fig. 4 is not very clear, how do you identify the sparse winning tickets and the most robust models (indicated using stars and triangles in the plots)?
- In the paper the authors consider also the case where IMP is applied in combination with well-known pre-training methods for low data regimes, such as SimCLR. However, it is not clear to me how the fine tuning is performed, IMP usually works when the weights are rewinded to initialization (or to an early epoch), therefore I'm not sure if (and how) it can be combined with pre-training techniques.

**Limitations:**

 The authors adequately addressed the limitations and potential negative societal impact of their work

**Strengths And Weaknesses:**

Strengths:
- The authors show an interesting application of lottery tickets.
- The experiments presented to validate their assumption are accurate and quite convincing, considering also synthetic noise and domain shifts. The authors consider also the case where lottery tickets are used in combination of other techniques for limited-data regimes, such as data augmentation.
- The paper is well written and easy to follow.
- I think that the Discussion section is very interesting, since it tries to give some insights on why lottery tickets can be a valid option in low data regimes. Since the paper is an empirical study, it is important to provide also an analysis that tries to explain the performance shown in the experiments and identifies the properties of the lottery tickets that are useful in this context.

Weaknesses:
- Since this paper is based on empirical evidence, I would have appreciated to see some experiments on different architectures. In the paper the authors perform all the experiments using ResNet-18, which is a quite simple architecture. I am curious to see if the same results could be obtained using also other architectures.
- I found the results not very surprising. Even though there are no prior works on sparse networks for data-limited regimes in the context of image classification, there are some results in NLP.

---

> ### Author Response · Authors · 2022-08-02
> **Response to Reviewer Skos**
>
>
> We thank the reviewer for the valuable comments and constructive suggestions. Below are our summarized questions and responses.
>
> **Q1: Inclusion of results beyond ResNet18**
>
> We evaluate pruning with several architectures in Tables. 1, 2, including the larger ResNet-50, VGG, MobileNet architectures. Our results indicate that sparse networks improve performance by almost **6%** (absolute) when compared to the corresponding dense networks. We shall also include results on non-convolutional-based architectures like VIT and discuss these results in the updated draft.
>
> **Q2: Clarification of identifying winning tickets and robust models in Fig. 4**
>
> *Figure 4: Robustness of models trained on CIFAR10 subsets on synthetic transforms (row-1) and domain-shifted (row-2) datasets. Shows performance of models trained on CIFAR10 subsets of different sizes (columns) at different sparsity levels (x-axis) for 4 augmentation strategies (lines). Sparse winning tickets identified on the CIFAR10 val. set are indicated by ⋆ and △ indicates most robust models.*
>
> The **winning tickets** in Fig. 4 are obtained based on the CIFAR10 validation set accuracy on the clean data (or directly corresponds to * in Fig. 2). The **most robust model** corresponds to a sparse network which achieves maximum robustness performance amongst the ones obtained at various pruning iterations during clean data training. Please note that we do not train (or prune) the network based on the corrupted data.
>
> **Q3: Pruning from Pretrained Weights**
>
> IMP from a pre-trained initialization is not expected to work too well [1]. We follow the experimental setup described in Sec. 3.3 where the network is initialized to the pre-trained weights and we rewind to the one after 2 epochs. We also adopt the strategy of using lower-learning rates to fine-tune from a pre-trained checkpoint.
>
> [1] Movement Pruning: Adaptive Sparsity by Fine-Tuning

---

### Official Review · Reviewer_KGu3 · 2022-07-10

**Rating:** 7
**Confidence:** 4
**Soundness:** 3 good
**Presentation:** 4 excellent
**Contribution:** 3 good

**Summary:**

The paper presents a thorough study of sparse wining tickets in the context of limited data settings. Using iterative magnitude pruning and rewinding, the authors train deep models with limited data, and show that sparse winning tickets outperform standard dense networks. The authors also show that augmentation helps improve the performance for the sparse tickets. In addition, the authors study a long-tailed classification setup and observe that sparse WT outperform dense models even when trained with auto-augment.

**Questions:**

1. Can the authors share the parameters for their augmentation setup?

**Limitations:**

The authors have mostly addressed the limitations by including experiments for a variety of setups.

**Strengths And Weaknesses:**

**Strengths**:

1. The paper makes an interesting observation on the few-shot learning capabilities of sparse models. In addition, the experiments mostly support their claims.
2. The experiments in Sec. 5 reveal some interesting properties of sparse wining tickets that aid in data-efficient learning.
3. Technically, the paper appears to be sound, with principled experiments and observations.

**Weaknesses**:

1. While not a weakness per se, it appears that the combination of IMP + augmentation is actually the winning combination for data-efficient learning. Especially, fig. 2 and fig. 4 both show that augmentation appears to play a larger role. Perhaps the authors could add some points to their discussion.
2. Another point of improvement could be training other architectures (transformers, VGG) to see if these properties are dependent on the architecture.
3. Liu et al [1] ,have also studied pruning for few shot learning. The authors should possibly include a citation and discussion.


Overall, the paper is a strong contribution by itself and reveals interesting properties.

[1] Liu et al., An Embarrassingly Simple Baseline to One-shot Learning, 2022

---

> ### Author Response · Authors · 2022-08-02
> **Response to Reviewer KGu3**
>
> We thank the reviewer for the valuable comments and constructive suggestions. Below are our summarized questions and responses.
>
> **Q1: The relevance of combining augmentation and pruning**
>
> It appears that augmentation plays a very important role in substantially improving the performance of our pruned networks. However, we wish to highlight that just augmenting the dense network does not lead to similar improvements and in fact, it is the combination of pruning and augmentation which leads to the best performance, as revealed consistently across our experiments (see Fig. 3(a), Tables. 1, 2). IMP prunes weights based on their magnitude - which is dependent on the quality of the network being trained i.e a network with poorly learned weights would be pruned inappropriately when compared to one with better. Therefore, we hypothesize that augmentations help the network learn better weights given a small data size which automatically helps mask irrelevant weights.
>
> | Dataset | D | WT | D + Aug* | WT+Aug* |
> |---|---|---|---|---|
> | CIFAR10 (2%) | 44.3 | 45.25 | 55.14 | 70.05 |
> | CIFAR10 (1%) | 36.28 | 38.31 | 43.8 | 59.66 |
> | CIFAR100 (2%) | 13.33 | 14.89 | 17.21 | 25.06 |
> | CIFAR100 (1%) | 8.12 | 9.76 | 11.21 | 16.44 |
>
> Table: Comparison of the Dense model (D), sparse winning ticket (WT), and the best performing dense and sparse models with augmentation (+Aug*) trained on varying CIFAR10, and CIFAR100 subsets.
>
> **Q2: Inclusion of results beyond ResNet18**
>
> We evaluate pruning with several architectures in Tables. 1, 2, including the larger ResNet-50, VGG, MobileNet architectures. Our results indicate that sparse networks improve performance by almost **6%** (absolute) when compared to the corresponding dense networks. We shall also include results on non-convolutional-based architectures like VIT and discuss these results in the updated draft.
>
> **Q3: Minor corrections**
>
> - Thank you for pointing us to [1] which briefly tests random-pruning in few-shot learning and leaves it for future investigation. We shall cite their work in our updated draft.
> - We shall include parameters for our augmentation setup in the Appendix on the updated draft. Upon acceptance, we shall also open-source the code for future work.
>
> [1] An Embarrassingly Simple Baseline to One-shot Learning

---

### Official Review · Reviewer_7i7B · 2022-07-10

**Rating:** 5
**Confidence:** 4
**Soundness:** 2 fair
**Presentation:** 3 good
**Contribution:** 3 good

**Summary:**

The paper investigates whether sparse winning tickets (obtained using Iterative Magnitude Pruning) improve image classification performance in low-data tasks (defined as 50-100 samples per class). The paper shows that sparse winning tickets outperform the full network by 10-15%, *but* this requires training using data augmentation. Without data augmentation, the winning tickets perform similarly to the dense network.

The paper’s contribution is empirical: it shows that sparse winning improves performance on tiny datasets.

**Questions:**

All my questions are related to missing explanations/analyses for the empirical results.

1. Why do winning tickets improve performance? Is there any relation between the number of model parameters and the size of the dataset (including data augmentation)?
2. Why is data augmentation required for sparse winning tickets to improve performance on very small datasets (last two subfigures from Figure 2)?
3. Why do sparse winning tickets preserve the filters? (Figure 7)
4. Why do sparse winning tickets preserve the residual connections (Figure 7)?
5. What’s the connection between the output norms and the generalization of the representations? (Figure 6)
6. Figures 3a/b/c show that training using data augmentation provides a 3x improvement on the winning ticket compared to the dense network (in the 1% data scenario). Why does this happen? In contrast, for over 10% of the training data, the improvement is similar for the WT and the dense network.

**Limitations:**

I don't see any negative social impact on their work.

The paper should clearly mention that the findings only apply to obtaining sparse winning tickets using Iterative Magnitude Pruning (IMP). Thus, all limitations of IMP also apply to the current work (e.g., IMP doesn't work well for pre-trained networks, and the presented experiments also show that the identified tickets on pre-trained networks provide small performance improvement).

**Strengths And Weaknesses:**

# Strengths
- The paper provides a **good empirical contribution** to the literature on sparse networks and learning from small datasets. The results raise further questions that the community might continue investigating. I find the paper to be **relevant to the conference**.
- The paper contains **extensive experiments**, investigating winning tickets in different low-data scenarios (standard classification, synthetic noise, synthetic transforms, domain shifts, highly imbalanced datasets). They show that winning tickets consistently improve performance when the network is trained from scratch.

# Weaknesses
- The paper’s core weakness is having a **very shallow analysis of the experimental results**. The paper has extensive experiments with useful results, but I found the provided analysis of the results to be weak. The authors provide little reasoning/explanations for their findings. Contribution 4 (line 60) is not well supported by the provided experiments and written analysis. The discussion section shows additional experiments and only briefly discusses the results without providing answers to any *why does this happen?* questions.
	- Section 5.1, and Figure 5, don’t answer why the network capacity and connectivity are important.
	- Section 5.2
		- The paper mentions: “Retaining more weights in the initial layers appears to allow the sparse models to keep the filters for detecting primitive features such as edges”. What does this result imply? Why do sparse winning tickets preserve the filters? Why don’t denser winning tickets preserve the filters? One hypothesis is that sparse tickets (on low-data) are simple functions (because they keep few parameters on the upper layers) of very well-extracted low-level features (because they keep most parameters on the first layers). Please discuss the findings if you want to keep Figure 7 in the discussion section.
		- “the identified sparse networks at lower data sizes have three peaks which correspond to the residual connection layers of each block”. What does this imply?
		- “the residual connections enable stronger gradients, and hence the maximum magnitude weights are expected to reside on the residual branch thereby increasing its density”. Why do you believe that large gradients imply large weight magnitude? There can be large gradients in both negative and positive directions (thus, the sum of the weights updates will be zero, and the weights don’t change).
		- “The ratio of norms are lower for the winning ticket suggesting perhaps the model learns more generalizable representations”. What’s the connection between the output norms and the generalization of the representations?
		- “In fact, the WT+Aug network showcases higher similarity to the network trained on 100% data when compared to the dense+Aug network. The above observations go in hand to further support our empirical analysis regarding the generalizability of the learned representations.” How does this observation support your empirical analysis on generalization? It’s not self-evident what you are implying.
	- **Potential solutions**: The paper would be significantly improved if the authors could explain their results more deeply. If not, then I suggest that the paper should claim *only* to identify an interesting phenomenon (i.e., sparse winning tickets outperform dense networks on small datasets). Also, the abstract/introduction/Contribution should clearly specify that the paper is purely empirical.

# Small changes
- In the title, replace “recognizers” with “classifiers” because the paper investigates only classification tasks
- line 33: what “inductive bias” does pruning provide?
- Improve citations. Many cited papers were published in conferences, but you’ve referenced the arxiv article. Please reference the published versions of the papers.
- In section 3.2, consider reordering the experiments: 1, 3, 2, 4, 5. I think 1 and 3 are related: 1 investigates data augmentation, and 3 investigates other training methods for small datasets (outside data augmentation). 2 and 4 also seem related because 2 analyses memorization and 4 investigates generalization.
- Consider moving section 3.5 (datasets) before section 3.4 (Augmentation strategies)
- Line 220: clarify what type of “memorization” are you referring to. As I understand, you test the “memorization” of the domains of the images because the experiments from Section 4.2 investigates synthetic transforms and domain shifts.
- Lines 235-237 provide an overstatement: “these results clearly demonstrate that winning tickets… are capable of avoiding memorization of training samples at the smaller data sizes, and much more so than the dense models”. There are many types of memorization, and the presented results don’t “clearly demonstrate” that winning tickets “avoid” it. Rephrase and reduce the overstatement.
- line 196 can be removed
- line 248 - add a short one-line about the argument [37] makes on why IMP doesn’t work well on pre-trained networks (i.e., because the magnitude of the weight doesn’t change much during fine-tuning, and the pre-training task largely determines the winning tickets)
- align tables 3 and 4
- Put figures in the order they are referenced in the paper. Figure 2 is referenced after Figure 3 .
- Move the title of Section 4.2 before the text starts (line 220)
- In line 316, add a short one-line description of how the CKA works

---

> ### Author Response · Authors · 2022-08-02
> **Response to Reviewer 7i7B**
>
> We thank the reviewer for their time and the detailed feedback. The questions and suggestions are helpful in further improving the quality of our work. Please find our point-by-point responses below and we would be happy to further discuss them here.
>
> **Q1: Clarification on the importance of Network Capacity and Connectivity**
>
> All the networks apart from D + Aug* are of lower capacity than the original network and as indicated by the results in Fig. 5, they showcase performance improvements - more drastically at the least data sizes (2%, 1%). This indicates that capacity indeed plays an important role in data efficiency. However, we also see that  WT+Aug* outperforms a network of a similar capacity Small Dense + Aug* indicating that beyond capacity perhaps the network connections also play an important role. To test this, we compare WT+Aug* against random connections at a network level (Random Prune + Aug*), layer level (Same Layer Sparse + Aug*) and observe that it outperforms both, indicating the relevance of a learned connection. Therefore, from these results, we could conclude both network capacity and connectivity play a vital role in improving the data efficiency of sparse networks.
>
> | Dataset | D+Aug* | Small D + Aug* | Random Prune + Aug* | Same Layer Sparse + Aug* | WT+Aug* |
> |---|---|---|---|---|---|
> | CIFAR10 (2%) | 55.14 | 59.69 | 55.23 | 61.58 | 70.05 |
> | CIFAR10 (1%) | 43.8 | 39.16 | 43.35 | 40.37 | 59.66 |
>
> **Q2: Clarifications on “Dense” Initial Layers**
>
> A popular understanding of convolutional networks is that the initial layers capture low-level information which is generalizable across datasets. Similarly, we observe that the winning ticket at lower data sizes automatically learns to retain dense initial layers which act as evidence for its improved performance. One hypothesis is that sparse tickets (on low-data) are simple functions (because they keep few parameters on the upper layers) of very well-extracted low-level features (because they keep most parameters on the first layers) which is consistent with our observations. If the reviewer can elaborate more on how our observations are lacking in answering the "why" we would like to understand and address it better. Specifically, if you have suggestions on how we can further improve our analysis we can try to incorporate them. It is also true that we do not make theoretical connections, if the reviewer can point to references in this regard, we would be happy to include them to help point readers in relevant directions.
>
> **Q3: Clarifications on “Dense” Residual Layers**
>
> From Fig. 7, we also see that the winning ticket tends to retain denser residual connections, and in Fig. 8, we notice that the winning ticket exhibits stronger feature similarity across layers. Therefore, it is possible that the increased layer-similarity is directly related to the dense residual connections rather than actually extracting globally generalizable features. To test this, we compute the l2 norm ratios of the outputs from the residual branch and main branch and we see in Fig. 6, that the ratios are surprisingly lower in the sparse network than in the dense network. This indicates that the residual branches of the sparse networks do not propagate as much magnitude of information as their corresponding dense counterparts indicating the higher layer-wise similarity has no correlation to the dense layers. This potentially suggests that the sparse network is in fact learning globally generalizable features. This is currently mentioned in different parts of the paper and we shall incorporate them in one place for better discussion.
>
> **Q4: Clarification on feature similarity between winning ticket trained on 1% data and dense network trained on 100% data**
>
> In Fig. 8, we compute the layer-wise CKA plots between the dense, winning tickets of a network trained only on 1% data to a network trained on 100% data. We can clearly see that the winning tickets showcase higher similarity over the dense network. This would not have been possible unless the winning ticket extracts generalizable features even in the absence of sufficient data.
>
> **Q5: Why do winning tickets improve performance?**
>
> We answer this question in the context of our work and with empirical evidence - why does winning tickets improve data-efficient performance? We identify that the lower capacity, and learned connections both at an overall network level and layer level contribute significantly to improved performance (Sec 5.1). Additionally, we discuss other observations including - the retention of dense initial layers to extract low-level features (Sec. 5.2), and the extraction of globally generalizable features (Sec 5.3, 5.4). We notice that lower network sizes (or parameter count) improve performance in smaller datasets (see Fig. 5) and the presence of augmentations always seems to help performance whether in the case of dense, or sparse networks.

---

> > ### Comment · Reviewer_7i7B · 2022-08-03
> > **The "Discussion" section merely presents results, doesn't *discuss* them**
> >
> > I thank the authors for their reply.
> >
> > The authors haven’t uploaded an updated manuscript version, so committed changes such as “improving discussion” aren’t present.
> >
> > The authors’ answers add *little* depth to the explanations provided in the paper. For example, I find the answer to Q5 “Why do winning tickets improve performance?” unsatisfactory (“We notice that lower network sizes … and the presence of augmentations always seems to help performance”).
> >
> > It’s ok if the authors can’t propose specific conjectures of why the presented phenomenon happens. Uncovering interesting phenomena is valuable for the community -- and I suggest the author start section 5 by clearly saying, “we perform an empirical investigation into the properties of the winning tickets and present interesting properties”.
> >
> > The current Section 5 is *not* a discussion. It simply presents results without discussing them (e.g., merely saying “we observe X” is not a discussion. For example, a discussion is “we observe X. We believe this is caused by Y. Because X happens, we believe this implies Z etc”). A discussion should include more input from the authors; the reader wants to know how you *interpret* the results.
> >
> > I maintain the paper's rating.

---

> ### Author Response · Authors · 2022-08-02
> **Response to Reviewer 7i7B [Contd.]**
>
> **Q6: Performance improvements upon augmentation during IMP.**
>
> We notice that combining augmentations always help improve performance both in the case of dense or sparse networks and across all data sizes. More surprisingly combing augmentation with pruning leads to major performance improvements by as much as 16% in the least data sizes (2%, 1%), while we do not see such large differences at data sizes like 20%, 10%. We reason that perhaps the 20% or 10% CIFAR10 subsets are not small enough where sparse networks can really shine. Regarding "why", we are not sure. If the reviewer has suggestions on a hypothesis and how we can verify them or suggestions on theoretical works that have looked at similar observations we would be happy to learn and see if we can incorporate them.
>
> **Con 1: Improve Discussion**
>
> We shall improve the clarity of text in Sec. 5 to discuss our observations and their corresponding inferences in greater detail.
>
> **Con 2: Highlight the ‘Empirical’ nature of our contributions**
>
> Thanks. We shall edit the text at the beginning of the paper to highlight that our contributions are empirically-driven.
>
> **Con 3: Minor Corrections**
>
> We shall incorporate a few of the suggested changes in our updated draft.
> - Improve citations to published versions.
> - Remove line 196, Include a one-line description of how CKA works.
> - Improve alignment of tables and figure re-ordering.

---

### Official Review · Reviewer_ED1y · 2022-07-11

**Rating:** 6
**Confidence:** 4
**Soundness:** 3 good
**Presentation:** 3 good
**Contribution:** 3 good

**Summary:**

This paper provides empirical evidence that sparse networks obtained using the lottery ticket hypothesis are also effective recognisers in settings where small datasets are available. Extensive experiments are conducted on multiple datasets and several interesting findings are reported, e.g., sparse networks in low data regimes are robust to synthetic noise, density is preserved in initial layers etc.

**Questions:**


See comments/questions in weakness section.

**Limitations:**

Does not apply to this paper.

**Strengths And Weaknesses:**

Strengths

The paper is well written.

Interesting findings are reported.

Extensive studies are conducted.


Weaknesses

Most of the experiments and findings are based on the CIFAR datasets. Why not using a more challenging dataset, like ImageNet? It would make the results more convincing.

Novelty is limited, existing methods are used, and extensive experiments are run (and the findings are definitely interesting) but it would be desirable if some novelty was included.

---

> ### Author Response · Authors · 2022-08-02
> **Response to Reviewer ED1y**
>
> We thank the reviewer for the valuable comments and constructive suggestions. Below are our summarized questions and responses.
>
> **Q1: Regarding experiments beyond CIFAR10.**
>
> In Appendix. D, we discuss results on more complex datasets - based on the number of classes including CIFAR100, and ImageNet. For CIFAR100, we create two data subsets of size - 2% (or 10 images/class) and 1% (or 5 images/class), while for ImageNet, we reuse the benchmark from [1] which uses a 5% (or 50 images/class) subset of the full dataset. From Table. 7, we can clearly see that the winning ticket outperforms the dense network by as much as **3%** in the ImageNet dataset and **5%** in the CIFAR100 dataset. This further justifies our overall claim that sparse networks improve data efficiency.
>
> | Dataset | D + Aug | WT + Aug |
> |---|---|---|
> | ImageNet (5%) | 28.82 | 31.04 |
> | CIFAR100 (2%) | 17.21 | 25.06 |
> | CIFAR100 (1%) | 11.21 | 16.44 |
>
> Table. 7: Comparison of the dense (D+Aug), and sparse winning ticket (WT+Aug) trained on complex data subsets with many classes.
>
> [1] Convolutional layers can exploit absolute spatial location.

---

> > ### Comment · Reviewer_ED1y · 2022-08-09
> > **Response**
> >
> > I would like to thank the authors for their reply.

---

### Author Response · Authors · 2022-08-02
**Overall Response**

We thank all reviewers for their thoughtful comments. The reviewers have all correctly summarized our study on iterative magnitude pruned networks (winning tickets) in the context of data-efficient classification tasks. Reviewers agree that the paper is well written (ED1y, 7i7B, Skos), includes extensive experiments (ED1y, 7i7B, Skos) and makes interesting observations (ED1y, KGu3, Skos) providing empirical evidence for the claims (7i7B, Skos). They have primarily requested additional clarifications which we have addressed. Some key points we wish to highlight from our responses are noted below.

**Experiments beyond CIFAR10 include CIFAR100 and ImageNet in the low data regime.**

Reviewers 1 and 3 correctly note that many of our studies are on the CIFAR10 dataset. However, we wish to highlight that in Appendix D and Table. 7 we also report results for CIFAR100 in the 1% setting and ImageNet in the 3% setting. Our key observations and claims continue to hold here. Mainly, the winning tickets obtained from iterative magnitude pruning outperform their dense counterparts.

**Experiments with different architectures including Resnet50, VGG and MobileNet**

Reviewers 3 and 4 requested experiments on additional architectures beyond ResNet18. We highlight here that these are already included in section 4.3 in Tables. 1, 2 of the paper where we compare methods on Resnet50, VGG as well as MobileNet. We can perhaps highlight it better in the experiment and discussion sections.

**Role of data augmentation**

It is true that augmentation helps in the low data setting and has been well studied (cite Cubuk et. al. and MixMatch). However, we wish to note that pruning and winning tickets obtained from IMP show benefits in the data-limited regime beyond that of data augmentation and the difference is more stark compared to dense counterparts that use the same augmentation strategies. We note more details in our responses.

We thank all reviewers for their feedback and comments on improving our analysis, discussion, and overall manuscript and are working to incorporate the changes.

---

> ### Author Response · Authors · 2022-08-08
> **Rebuttal Revision**
>
> We have updated our draft to clarify and include many of the changes requested by the reviewers (indicated in blue). We apologize for the delay and once again thank the reviewers for their time and feedback. We would be happy to provide any further clarifications.

---

### Meta-Review · Area_Chair_YoCh · 2022-08-24

**Recommendation:** Accept
**Confidence:** Less certain

**Metareview:**

The paper has received borderline and positive reviews. Overall, the reviewers find the empirical contribution of the paper to be interesting and solid enough (even though one reviewer find the explanations given in the paper to be a bit shallow). The rebuttal was nevertheless convincing. The area chair agrees with the reviewers' assessment and follows their recommendation.

**Award:**

No

---

### Decision · Program_Chairs · 2022-09-14

Accept